# Insights into Circulating Tumor Cell Clusters: A Barometer for Treatment Effects and Prognosis for Prostate Cancer Patients

**DOI:** 10.3390/cancers14163985

**Published:** 2022-08-18

**Authors:** Linyao Lu, Wei Hu, Bingli Liu, Tao Yang

**Affiliations:** 1Center for Medical Research and Innovation, Shanghai Pudong Hospital, Fudan University Pudong Medical Center, Shanghai 201399, China; 2Department of Urology, Shanghai Pudong Hospital, Fudan University Pudong Medical Center, Shanghai 201399, China; 3Department of Orthopedics, Shanghai Pudong New Area People’s Hospital, Shanghai 201299, China

**Keywords:** circulating tumor cells, circulating tumor cell clusters, prostate cancer, epithelial–mesenchymal transition, metastasis, androgen receptor V7

## Abstract

**Simple Summary:**

Circulating tumor cells (CTCs) are a promising biomarker for the risk of prostate cancer aggressiveness and metastasis and play a role in the processes of tumor migration and metastasis. CTC clusters, which have different physical and biological properties from individual CTCs, are collections of tumor cells and non-malignant cells, resulting in greater metastatic potential. Therefore, this review aims to summarize the current knowledge of CTC clusters in metastasis as well as related biological properties and to suggest possibilities for their usage in diagnostic and therapeutic practice.

**Abstract:**

Prostate cancer (PCa) exhibits high cellular heterogeneity across patients. Therefore, there is an urgent need for more real-time and accurate detection methods, in both prognosis and treatment in clinical settings. Circulating tumor cell (CTC) clusters, a population of tumor cells and non-malignant cells in the blood of patients with tumors, are a promising non-invasive tool for screening PCa progression and identifying potential benefit groups. CTC clusters are associated with tumor metastasis and possess stem-like characteristics, which are likely attributable to epithelial–mesenchymal transition (EMT). Additionally, these biological properties of CTC clusters, particularly androgen receptor V7, have indicated the potential to reflect curative effects, guide treatment modalities, and predict prognosis in PCa patients. Here, we discuss the role of CTC clusters in the mechanisms underlying PCa metastasis and clinical applications, with the aim of informing more appropriate clinical decisions, and ultimately, improving the overall survival of PCa patients.

## 1. Introduction

In the United States, prostate cancer (PCa) is the most common cancer diagnosed in males [1,2,3]. Since prostate-specific antigen (PSA) was widely applied to the detection of asymptomatic PCa during the early 1990s [4], overall PCa incidence in males has generally decreased. This has revealed the immense benefit of inspection tools. Recently, however, it has been found that early detection of PCa by PSA testing has led to overdiagnosis and overtreatment. Thus, there is an urgent need for alternative tools [5]. Liquid biopsy, defined as the analysis of tumor cells and tumor-derived products in the blood and other body fluids [6], is an alternative to tissue biopsies. It can be used to diagnose and screen for tumors in real-time. Additionally, it is a noninvasive and replicable way of monitoring circulating tumor cells (CTCs), cell-free DNA (cfDNA), cell-free RNA (cfRNA), and extracellular vesicles and particles (EVPs). Studies have demonstrated that CTCs’ diagnostic efficiency is superior to that of PSA when patients’ PSA levels are between 4–10 ng/mL [7]. In conclusion, biomarkers and enumeration of CTCs show the prospective evaluation of prognosis and treatment efficacy in metastatic PCa [8,9,10].

CTCs are rare in the peripheral blood. For instance, single CTCs’ prevalence is from 1 to 10 CTCs per 10^6^–10^8^ white blood cells [11], and CTC clusters are estimated to constitute only 2–5% of all CTC events detected in circulation [12]. The characteristic markers of CTCs for isolation include epithelial cell adhesion molecules (EpCAM; transmembrane glycoprotein), vimentin (VIM; structural cytoskeletal protein), and cytokeratins (CK8, 18, 19). According to biomarkers and the physical characteristics of CTCs, several recent methods have been designed to enrich and isolate CTCs from blood cells (especially white blood cells) (Table 1). For example, devices have been developed to isolate and detect CTCs based on the presence of specific proteins (CellSearch^®^(Menarini Silicon Biosystems, Bologna, Italy), CTC-chip, RosetteSep), gene transcripts (AdnaTest), size (microfluidic chips), density (Oncoquick), electric charge, secretion of specific proteins (EPISPOT) and invasive properties [13]. To date, CTCs have been reported in various solid tumors including breast cancer [14], prostate cancer, lung cancer [15], colon cancer [16], liver cancer [17], and head and neck cancer [18]. CTCs are generally referred to as either single CTCs or CTC clusters, with the latter considered to have a 23- to 50-fold greater metastatic potential [12,19]. Studies suggest that a CTC cluster contains at least 2 tumor cells (and up to 100) and several non-malignant cells including but not limited to a heterogeneous group of cells, for example, tumor-associated macrophages (TAMs), cancer-associated fibroblasts (CAFs), white blood cells, epithelial cells and platelets [20,21,22,23,24,25]. In contrast to single CTCs, CTC clusters show distinctive phenotypes, gene expression, and metastasis patterns, indicating unique biological properties in neoplasm metastasis. CTC clusters can be detected in men with either localized PCa or metastatic PCa, and there is a larger enumeration of CTC clusters in men with advanced PCa during multiple stages of cancer recurrence and metastasis [22,26,27,28,29]. Moreover, CTC clusters have also been detected in prostate cancer patient-derived xenograft (PDX) models. This has provided a new tool for exploring PCa metastasis [30].

Many questions still remain about CTC clusters. For instance, is the direct derivation of CTC clusters from primary tumors or single CTCs in the peripheral blood? Likewise, the relationship and interaction between CTC clusters and single CTCs remain unclear. How do CTC clusters metastasize? How are CTC clusters related to stem cells? In this context, these topics have been reviewed with an emphasis on the relationship between CTC clusters and PCa metastasis, as well as the prospective application of CTC clusters in clinical settings.

## 2. CTC Clusters and Single CTCs

To date, despite there being no solid evidence that CTC clusters and single CTCs are two totally different and independent cells, the contrast between CTC clusters and single CTCs based on physical properties and biological features is reported in many tumors. In patients with tumors, single CTCs are more prevalent than CTC clusters, and CTC clusters consisting of several tumor cells are larger than single CTCs. This lessens the potential for extravasation. However, CTC clusters have been observed reversibly unfolding into single chains when they go through vessels [34]. Non-malignant cells in CTC clusters are involved in extravasation and stabilize CTC clusters in peripheral blood [35]. In addition, single CTCs are unable to form polyclonal metastatic foci in distant organs, but CTC clusters have a greater ability to metastasize and also have the potential to form polyclonal metastatic foci [36]. In conclusion, the composition, survival advantage, and metastatic potential are the main differences between CTC clusters and single CTCs.

## 3. CTC Clusters and EMT

### 3.1. Locations of Cells with Different E/M States in CTC Clusters and Invasion

EMT is a complex cellular pathway in which epithelial cells lose epithelial characteristics (e.g., cell-to-cell adhesion) and gain mesenchymal characteristics (e.g., increased migratory capabilities) [37]. Experimental evidence that has accumulated over decades has indicated that tumor cells in CTC clusters undergo EMT, as demonstrated by EMT biomarker detection [30,38,39]. The evidence also reveals that EMT has potential relevance to mechanisms underlying tumor metastasis, cancer stem cell (CSC) generation and maintenance, as well as drug resistance [40]. At the molecular level, the loss of adherens junction protein E-cadherin is considered a hallmark of EMT [41]. It results in the gain of mesenchymal markers such as vimentin, N-cadherin, α-smooth muscle actin (α-SMA), and fibronectin [42]. Satelli et al. reported that FOXC2, an EMT-specific marker, was detectable in CTCs from 10 metastatic PCa patients. However, the epithelial markers EpCAM and E-cadherin were absent in these cells, indicating a mesenchymal phenotype [32]. Surprisingly, Yu et al. found an association between the expression of mesenchymal markers and CTC clusters in human breast cancer specimens, rather than single migratory cells [43]. This has focused following researches on CTC clusters and EMT. In PCa-based PDX models, CTC clusters have been observed to contain a mix of cell phenotypes, and the location patterns of different cell phenotypes are worthy of inquiry. For instance, epithelial-like cells have been located on the periphery of the cluster, surrounding hybrid or mesenchymal-like cells in PCa-based PDX models [30]. The epithelial phenotype has been implicated in metastatic colonization [40]. Lori E. et al. demonstrated that PCa with an increasingly mesenchymal phenotype shed greater numbers of CTCs more quickly and with greater metastatic capacity than PCa with an epithelial phenotype in 4 PCa models with progressive epithelial (LNCaP, LNCaP-C42B) to mesenchymal (PC-3, PC-3M) phenotypes [44]. Thus, epithelial-like cells on the periphery of CTC clusters might be essential to understanding mechanisms underlying distant metastasis. However, tumor cells with an EMT phenotype have exhibited a reverse location pattern in solid tumors. In primary and secondary PCa, the expression of the EMT phenotype at the invasive tumor front is higher than that at the center [45]. A similar phenomenon has also been observed in other solid tumors such as breast cancer [46]. Thus, it is the different environments that might result in some variation in cell phenotypes in CTC clusters or solid tumors. Additionally, it is unclear whether the distribution of CSC-like cells might have differential effects on cell phenotype dedifferentiation or conversion within CTC clusters or solid tumors. Collective cell migration has been observed not only in tumors but also in wound healing and tissue renewal. Cells in collective migration can perceive the microenvironmental chemotaxis and initiate the cellular migration by dividing into “leader” cells and “follower” cells [47]. Single-cell RNA-sequencing analysis of leader-like and follower-like cells has revealed differentially expressed gene profiling pertaining to cellular locations within the migrating collective [48]. For example, genes associated with Wnt/planar cell polarity (PCP) signaling were overexpressed in cells at the invasive front. This indicates that Wnt/PCP signaling is involved in tumor invasion [48,49]. Luo et al. demonstrated that the crosstalk between androgen receptor (AR) and Wnt signaling promotes the androgen-independent growth of PCa by maintaining LNCaP cells under androgen-depleted conditions. *WNT5A* and *LEF1* are reported to be downregulated in low-grade PCa while upregulated in metastatic PCa [50]. In addition, a retrospective analysis suggested that non-canonical Wnt signaling was activated in CTCs of 13 PCa patients with AR inhibitor, resulting in antiandrogen resistance [51]. In PCa, ectopic expression of Wnt5a suppresses the anti-proliferative effect of the inhibition of AR, whereas this suppression restores partial sensitivity in drug-resistant cells. Wnt5a, a vertebrate Wnt ligand, triggers the Wnt/PCP signaling pathway. Ultimately, it regulates cytoskeletal remodeling, such as cell polarity, migration, and subsequently, tissue re-arrangement and organ formation [52]. The assessment of noncanonical Wnt signaling pathway components seems to be a promising way to identify patients with metastatic castration-resistant prostate cancer (mCRPC). These patients are likely to have poor prognoses after treatment with androgen deprivation therapy (ADT). Therefore, the interaction between the Wnt signaling pathway and the AR signaling pathway might be relevant to the ADT resistance. In patients’ CTC clusters, the distribution of tumor cells in different E/M states might imply that tumor cells perform various roles in the migration process. 

### 3.2. Stem-Like State of Hybrid E/M Phenotype Cells and Metastasis

It has been assumed that many intermediates between epithelial and mesenchymal phenotypes co-exist in the migrating CTC cluster. The hybrid E/M phenotype cells form heterogeneous CTC clusters with cells in various EMT states and maintain cell–cell junctions on the basis of E-cadherin [53,54]. However, the function of each phenotype remains controversial. Several recent studies have suggested that it is hybrid E/M cells that act as pluripotent CSCs and promote invasiveness [42]. To date, there have been consistent observations in hybrid E/M cells isolated from mouse PDX models of PCa [55], as well as in the co-culturing of epithelial PCa cells with post-EMT PCa cells in vitro [56]. Further, this hypothesis has received the support of clinical evidence [57], which suggests that hybrid phenotypes “gain” features of both epithelial and mesenchymal phenotypes in carcinoma specimens [58,59]. Ruscetti et al. demonstrated that mesenchymal and epithelial states in PCa cells of *Cre^+/−^*; *Pten^L/L^*; *Kras^G/+^*; *Vim-GFP* mouse models contribute differentially to their capacities for tumor initiation and metastatic seeding, respectively [55]. Mesenchymal tumor cells display an enriched tumor-initiating capacity, and epithelial tumor cells can exist with the capacity to form macro-metastases. A hybrid phenotype that possesses both properties of mesenchymal and epithelial tumor cells has an enhanced ability to migrate and form distant foci in a complex environment (Figure 1). However, some researchers have held opposing views. Tsuji et al. demonstrated that only appropriate cooperation between EMT cells and non-EMT cells of HCPC-1 cells can promote successful metastasis; each kind of cell alone is unable to metastasize [60]. In this study, the researchers did not consider the hybrid E/M state. Furthermore, several findings have indicated that EMT is dispensable for cancer cell-mediated migration. For instance, Fischer et al. demonstrated that inhibiting EMT by targeting ZEB1 and ZEB2 overexpression via miR-200 does not impair breast tumor cells’ ability to form distant lung metastases [61]. Likewise, Zheng et al. suggested that EMT inhibition by deleting Snail or Twist is critical to neither robust invasion nor the metastasis of pancreatic cancer [62]. Together, CTC clusters undergoing EMT are pivotal for migration, but hybrid E/M phenotype cells’ exact role in metastasis is worth pursuing further.

The hybrid EMT cell state, with both epithelial and mesenchymal features, is reconcilable with the state of highly plastic stem-like cells. Similarly, in the process of blood dissemination, CTCs undergoing EMT possess features of CSC-like tumor cells that are responsible for generating most of the metastatic foci, as well as capabilities of resistance to radio- and chemotherapy-based treatment [11,63]. Likewise, CTC clusters have been observed expressing more mesenchymal transcripts in patients receiving cancer treatment [64]. A study suggested that 35/46 (76%) CTCs were CD133-positive, a putative prostate cancer stem cell marker, in 35 patients with high-risk, localized prostate cancer; the researchers demonstrated that the CD133 and E-cadherin-positive CTC fragments were associated with biochemical recurrence at 1 year [65]. Compared to single CTCs in the DNA methylation landscape of 43 breast cancer patients and 3 mouse models, CTC clusters resulted in hypomethylation in the binding sites of OCT4, NANOG, SOX2, and SIN3A. However, single CTCs featured hypomethylation of other TFBSs, including those that are occupied by MEF2C, JUN, MIXL1, and SHOX2 [19]. Additionally, most of these binding sites were occupied by master stemness and associated with proliferation regulators. The patterns were similar to those of embryonic stem cells. Of note, NANOG is essential to the establishment of pluripotency, self-renewal, and reprogramming [66], which regulate the gene expressions involved in the mitochondrial metabolic pathways required to maintain tumor-initiating stem-like cells in PCa and breast cancer [67]. Tumor heterogeneity, which is generated evolutionarily not only as a result of genetic alterations but also by the presence of cancer stem cells, is a driving factor behind the failure of cancer treatment modalities. Therefore, elucidating the molecular underpinnings of CSCs’ biological features is crucial to the development of novel cancer therapies for PCa.

## 4. CTC Clusters and Metastasis

### 4.1. Sources of CTC Cluster

It has been demonstrated that CTC clusters can reduce CTC apoptosis, elevate cell viability, and promote the ability to re-form clusters [12,68]. Furthermore, CTC clusters have been found to exhibit stronger resistance to anti-tumor drugs than single CTCs [69,70]. However, how CTC clusters form remains controversial. To date, there are two main hypotheses explaining how CTC clusters form (Figure 2). The first is that as tumor pieces, CTC clusters fall off the primary tumors and then travel through the vessels. The other is that CTC clusters are aggregated from single CTCs in the blood. Alone, CTC clusters are considered to be a part of the primary tumor, shedding into the blood spontaneously or passively. However, recent findings have challenged this hypothesis. Liu et al. have observed that individual CTCs derived from patient-derived breast cancer models aggregate into CTC clusters in the blood vessels, suggesting the potential resource of CTC clusters. Most of the clusters contain two types of tumor cells as labeled by fluorescent indicators, resulting in a high ratio of polyclonal CTC aggregation within 2 h in the lungs. However, the ratios of dual-color aggregates in the lungs gradually decrease over time [71]. This study provides evidence that CTC clusters are directly formed by individual CTCs in the peripheral blood. Then, what is the relationship between metastatic foci and CTC clusters? Maddipati et al. reported that most metastatic foci in the distant organs were polyclonal populations in a mouse model of pancreatic cancer; whereas metastatic foci became increasingly dominated by a single clonal population as they increased in size (Figure 2) [36]. In addition, genes that are highly expressed in primary tumors compared to CTCs are consistent with genes expressed in primary tumors compared to metastases. Furthermore, it has been reported that single CTC genomic profiling displays high concordance with metastatic biopsies from the same patients [72]. These studies imply that CTCs might be an origin of metastatic foci, and there is a differentiation between primary tumors and CTCs [51]. Cheung et al. determined that polyclonal lung metastases arise via colonization by a multicellular cluster of tumor cells instead of the serial seeding of single tumor cells. Additionally, CTC clusters were reported to exist in five different stages of metastasis: collective invasion, locally disseminated clusters in the adjacent stroma, intravasation tumor emboli, CTC clusters, and distant metastases [68]. Taken together, these observations suggest CTC clusters might form when there are a variety of heterogeneous tumor cells in the vasculature, and then gradually convert from polyclonal to oligoclonal foci after colonizing distant tissues. If CTC clusters are gathered by single CTCs, then how do individual CTCs recognize each other and maintain a state of aggregation?

### 4.2. Role of CTC Cluster Components

CTC clusters consist of not only tumor cells, but also non-malignant cells, including TAMs, CAFs, immune cells, epithelial cells, and platelets. In CTC clusters, non-malignant cells act as protectors and supporters for cancer cells in the blood and increase cancer cells’ ability to migrate and invade (Figure 1). In solid tumors, CAFs promote tumor invasion by multiple tumor types with multiple mechanisms, ranging from typical cell–cell signaling to dynamic alteration of ECM, which is the result of fibroblast remodeling activity. Likewise, PCa patients’ CTC clusters go through a high magnitude of fluid shear stress (FSS) in the peripheral blood, during which process reactive CAFs can conserve tumor cells’ proliferative capability by cell–cell contact and secreting paracrine factors [24]. The reactive CAF phenotype emerges from normal fibroblasts (NFs), which are activated by cytokines secreted by tumors. Additionally, reactive CAFs can be identified by the overexpression of α-SMA, fibroblast specific protein 1, and fibroblast activation protein [73]. Further, Duda et al. demonstrated that CAFs spontaneously spread to the lung tissue along with metastatic cancer cells and accelerate the growth of secondary tumors [74]. This evidence indicates that CAFs are particularly relevant in CTC clusters with metastasis as well as invasion. CAFs are one of the abundant stromal cell populations in the tumor microenvironments. In primary tumors, CAFs facilitate a breach of the basement membrane (BM) by altering the cellular organization and the physical properties of BM, making it less compact for tumor cell invasion [75]. Tumor protrusion and processes can be inhibited when tumor cells are exposed to healthy extracellular matrixes, indicating CAFs’ critical role in promoting metastasis [76]. Recently, it has been reported that a heterophilic adhesion among CAFs and tumor cells via N-cadherin or E-cadherin guides tumor cell migration within connective tissue [77]. That is in line with a previous report demonstrating N-cadherin- and E-cadherin-mediated cancer cell migration by cell–cell junction [78]. In sum, CAFs appear to be involved in cancer cell migration and invasion, especially in intercellular adhesion. In PCa, TAMs uniquely co-isolated with CTCs have been shown to promote conversion from epithelial–mesenchymal plasticity, resulting in the adaptation of cancer cells to mechanical stress. Thus, CTC clusters are conferred with adaptive resistance to shear stress and maintain integrity [20]. Additionally, platelets can act as a coating of CTCs to protect them from violent FSS, and their adhesive proteins (fibronectin and von Willebrand factor) have been found to interact with CTCs through integrins, supporting CTC cluster formation [78]. In addition, platelets have been shown to protect CTCs from apoptosis, promote EMT and extravasation, and facilitate escape from immune system surveillance such as by inhibiting natural killer (NK)-cell-induced lysis in PCa [79,80]. For instance, Labelle et al. demonstrated that platelet-derived TGFβ and the contact between platelets and tumor cells synergistically activate the TGFβ/Smad and NF-ĸB pathways in colon carcinoma and breast carcinoma cells. This results in the transition to an invasive mesenchymal-like phenotype [81]. Moreover, various immune cells also protect CTC clusters from anti-tumor immune attacks, promoting tumor cell migration. Szczerba et al. observed CTC–white blood cell clusters in breast cancer by staining EpCAM, human epidermal growth factor receptor 2 (HER2), epidermal growth factor receptor (EGFR), and CD45 [82]. Furthermore, neutrophils have also exhibited promotive effects on CTC migration. In conclusion, non-malignant cells within CTC clusters play a critical role in promoting extravasation by altering cell–cell junctions and offering protection from anti-cancer immune attacks and FSS in the process of cancer cell migration and colonization. 

### 4.3. Distant Metastatic Foci and CTC Clusters

Considering that CTCs can grow into emboli, researchers previously assumed that CTC clusters were unable to transit through capillaries 5–10 µm in diameter in view of cluster size. However, Au et al. recently presented evidence that over 90% of CTC clusters containing up to 20 cells successfully traverse 5- to 10-μm porous constrictions during detection by means of microfluidic devices. It has been noted that CTC clusters can rapidly and reversibly unfold into single chains by selective cleavage of intercellular adhesions when CTC clusters transit through narrow blood vessels (Figure 2) [34]. Similarly, Green et al. utilized a microfluidic device (a Pillar device and an X-magnetic device) to isolate single CTCs and clusters from whole blood in mouse models and patients with metastatic breast cancer. Surprisingly, they observed that some of the clusters could maintain weak intercellular adhesion. These clusters re-arranged their shape to travel through the pillars, and then clusters re-formed in the X-magnetic device [83]. This process provided direct evidence that CTC cluster cohesion can be regulated by the dynamic expression of E-cadherin, as reflected by the adaptive alteration of cluster shapes. Additional evidence revealing the metastatic potential of CTC clusters has been based on Na^+^/K^+^ ATPase inhibitors, which lead to DNA methylation remodeling at critical sites and metastasis suppression by dissociating CTC clusters into single cells [19]. These studies provide novel ideas on how CTC clusters transit to distant metastatic foci. CTC clusters exhibit much higher intercellular adhesion levels than single CTCs, as evidenced by a report that CTC clusters in breast cancer overexpress plakoglobin (by 219-fold), an important component of desmosomes and adherence junctions, in comparison to single CTCs [12]. In a CHD1-normal cohort of PCa patients, high junction plakoglobin expression, which was linked to strong androgen receptor expression, high cell proliferation, and *PTEN* and *FOXP1* deletion, was an independent predictor of poor prognosis and early biochemical recurrence [84]. Now, the precise role of intercellular adhesion in CTC cluster migration is still not well understood. Thus, identifying intercellular adhesion is crucial for elucidating the mechanism underlying PCa metastasis. 

## 5. Separation Techniques and Devices

Despite the growing number of methods associating biological attributes and physical properties, particularly EpCAM and size, with the isolation and detection of CTC clusters, it is still a major challenge to isolate CTC clusters. The current CTC cluster enrichment technologies mainly use biological or physical properties of the CTC clusters for isolation. The CellSearch^®^ system was the first and only US Food and Drug Administration (FDA)-authorized system for the enumeration of CTCs in 7.5 mL of blood to rely on the expression of EpCAM only [34], resulting in overlooking the more invasive EpCAM-negative CTCs that are in the process of epithelial–mesenchymal transition (EMT) [20]. Of note, some studies have suggested that non-blood-derived aneuploid circulating tumor-derived endothelial cells (CTECs) have properties in common with CTCs, such as the expression of EpCAM. Thus, CTECs’ influence on CTC detection is deserving of much attention [15]. Recently, to purify CTC clusters out of individual CTCs, researchers have utilized the physical and biological properties in combination, including the size, deformability, and expression of cellular markers [35,36]. This is a great opportunity to understand CTC clusters.

Microfluidic devices, which reduce cluster separation, accelerate processing time, and collect live CTC clusters for downstream analysis, display the potential to be the most favorable platform for isolating CTC clusters [85]. A number of intrinsic biomarkers have been used to identify and isolate CTC clusters: for example, size, electrical polarizability, and hydrodynamics in microfluidic systems [86]. For instance, due to the strong size-dependence of inertial migration, CTC clusters and single CTCs can be selected by precisely controlling the channel length in untreated whole blood [87]. Additionally, deterministic lateral displacement has been used to continuously separate CTC clusters by size. CTC clusters, having greater sizes than individual CTCs, are removed from whole blood by deterministic lateral displacement of the microfluidic device. CTC clusters are collected through successful deflection, while the remaining parts are sorted by discriminating asymmetric clusters from symmetric single cells. Au et al. [88] demonstrated that the recovery of cultured breast cancer CTC clusters from whole blood using this integrated two-stage device results in minimal cluster separation, 99% recovery of large clusters, and cell survival rates exceeding 87%. Moreover, the ability of electric fields to exert forces on particles has been used as a means of manipulation in stand-alone separation devices. Chiu et al. [89] reported that optically induced dielectrophoresis-based cell manipulation in a microfluidic system was able to isolate H209 small cell lung cancer cell clusters with a cell purity and recovery rate of 91.5 ± 5.6%, and 70.5 ± 5.2%, respectively, without compromising integrity. However, electrochemical reactions can result in free radicals, leading to significant cellular damage. Microstructural protrusions have been shown to be useful for entrapping cells and performing cell separation based on size, deformability, and density. Herringbone grooves form a flow pattern suitable for separating particles of similar size based on density. The Herringbone-Chip with low shear flow properties revealed the presence of CTC clusters in patients with metastatic prostate cancer [90]. In sum, microfluidic chips exhibit the ability of efficient cell sorting without purification, high system throughputs, and low sample volume.

## 6. CTCs in Clinical Application

Liquid biopsy is the process of collecting information relevant to disease in body fluid (e.g., blood), including CTCs, cfDNA, cfRNA, and EVPs. In PCa, liquid biopsy has the advantages of easy acquisition, high replicability, and superior reflection of cellular properties, treatment efficacy, and cancer prognosis. Regarding PCa patients with bone metastasis, liquid biopsy is superior to tissue biopsy, since the latter is technically challenging. In addition, in bone metastasis, disseminated tumor cells (DTCs) are a special kind of CTC that can home to the bone marrow. The amount of DTCs isolated from bone marrow is larger than that of CTCs isolated from blood [91,92]. During this process, DTCs are also a candidate for tumor sampling. From another perspective, cfDNA, also called circulating tumor DNA, is a DNA fragment released into the blood circulation from apoptotic, necrotic or secreted tumor cells. The genetic variation and epigenetic features of tumor cells can be well preserved in cfDNA, making it a potential biomarker [93,94,95]. However, the amount of cfDNA is hard to quantify due to the high degree of fragmentation and low circulating concentration. Moreover, cfDNA measurement can be complicated by heart dysfunction associated with heavy smoking or physical exercise. Alternatively, EVPs can also serve as an indicator of cancer treatment efficacy, and even for early diagnosis and prognosis estimation [96,97,98]. Hoshino et al. investigated EVP proteomic profiles in 120 human samples and demonstrated that plasma-derived EVPs can be utilized to detect early oncogenesis with 95% sensitivity and 90% specificity [99]. In conclusion, liquid biopsy (especially CTCs), as an emerging non-invasive testing method, plays a vital role in aiding diagnosis, assessing efficacy and prognosis, and monitoring early recurrence and drug resistance. As such, it may become an important component of future cancer management.

### 6.1. CTC Enumeration

For PCa, extensive findings have suggested that CTCs play a pivotal role in guiding chemotherapeutics [100,101], predicting prognosis [102,103,104,105,106,107], and reflecting treatment effects [108,109,110] (Table 2). With regard to the limited technological methods for purifying single CTCs and CTC clusters, most studies involving CTCs do not distinguish CTC clusters from single CTCs when they explore the role of CTC counting. For instance, several studies have established the prognostic role of CTC enumeration in both localized and metastatic PCa patients, demonstrating a higher prognostic performance. Choi et al. demonstrated that localized PCa patients who were CTC-positive were at a higher risk of up-staging and up-grading [111]. However, Meyer et al. suggested that there is no significant correlation between CTC detection and PSA, disease characteristics, or the development of biochemical recurrence in patients with localized PCa [112]. Zapatero et al. also did not observe any association between CTC count and OS in 65 patients with advanced high-risk localized PCa [113]. In sum, the effect of CTC counts on localized PCa has been controversial. Therefore, larger-scale trials with more sensitive techniques are needed to confirm it, and in metastatic PCa patients, the prognostic role of CTC enumeration has more practical applicability. Lozano et al. demonstrated that baseline CTC enumeration of 80 mCRPC patients treated with docetaxel had greater significance than PSA quantitation in predicting overall survival [114]. Kruijff et al. summarized 114 mCRPC patients treated with cabazitaxel in the second round of chemotherapy. They determined that the CTC count has a strong prognostic value, for both progression-free survival (PFS) and overall survival (OS) [115]. In addition, Goldkorn et al. conducted a phase 3 prospective randomized trial in metastatic castrate-sensitive prostate cancer (mCSPC) patients treated with ADT combined with orteronel or bicalutamide. After adjusting for disease extent and distinguishing between patients likely to experience poorer survival outcomes, they found that baseline CTC counts in mCSPC had a high predictive value for 7-month PSA and 2-year PFS (N = 264 and N = 336, respectively) [116]. In sum, the CTC count is a promising predictor of outcomes in patients with mCRPC. Researchers have further differentiated PCa patients by CTC cutpoints of 0, 1–4, and ≥5, hoping for a more reasonable prognostic prediction. A pilot study suggested that CTC-positive (≥1 CTC/7.5 mL blood) patients with recurrent PCa have worse pathological and short-term oncological outcomes [117]. In addition, Yang et al. demonstrated that total CTC count ≥5/5 mL blood was an independent predictor of early progression to mCRPC and shorter cancer-specific survival after cytoreductive radical prostatectomy for 54 oligometastatic hormone-sensitive PCa patients [118]. These findings indicate that CTC enumeration, as an early treatment response biomarker, is helpful in supporting the utilization of CTC quantification to identify patient cases with early progression. It is also useful in selecting intensive treatments to prevent unnecessary exposure to ineffective therapy with undesirable toxicities [114]. Taken together, there is extensive evidence of an association between CTC count, prognostic prediction, and therapeutic efficacy in PCa patients, especially those with metastatic PCa.

To date, increasing CTC-involved experiments have provided insight regarding both single CTCs and clusters. Recently, Zhu et al. observed that CTCs exhibited random bursts during cancer progression in an orthotopic mouse model of human PCa. The bursting activity was more intense in early stages and declined in late stages in PCa. CTC counts have been found to peak during night time, suggesting that the release of single CTCs might be regulated by circadian rhythms [119]. Thus, for the purposes of CTC detection, blood sampling from patients at appropriate timepoints should be considered.

### 6.2. Androgen Receptor V7

Androgen receptor V7 (AR-V7) in CTCs has been studied widely for PCa as well (Table 3), and it has been found that the expression of AR-V7 is associated with cancer prognosis and drug resistance (Figure 3). AR-V7 is a truncated isoform of the canonical AR-FL protein, and it lacks the ligand-binding domain, which is the target of enzalutamide and retains both the DNA-binding domain and the amino-terminal domain [120]. Areti et al. suggested that *AR-V7* detected in 34/69 (49.3%) CTCs is superior to that detected in 4/52 (7.7%) plasma-derived EVP as an indicator of cancer treatment efficacy in metastatic PCa patients [121]. Moreover, Sepe et al. confirmed that AR-V7 expression had a significant negative effect on radiological OS in 37 mCRPC patients treated with enzalutamide or abiraterone [122]. The detection of AR-V7 in CTCs is associated with shorter PFS and OS in mCRPC patients treated with abiraterone or enzalutamide, but such AR-V7-positive patients still derive similar benefits from subsequent taxane chemotherapy [101]. In a cross-sectional cohort study, Scher et al. found that 161 mCRPC patients who had pretherapy AR-V7-positive CTCs treated with taxanes had a more favorable survival time with taxanes relative to AR signaling (ARS) inhibitor. Additionally, it was demonstrated that cabazitaxel improves overall OS in mCRPC patients after docetaxel [123]. Gurioli et al. suggested that the initial reduced dose of cabazitaxel (20 mg/sqm) had a correlation between AR-V7 expressions and worse outcomes in eight mCRPC patients [102]. In sum, the detection of AR-V7 in CTCs is associated with poor outcomes in abiraterone-treated patients and enzalutamide-treated patients, but the presence of AR-V7 can be a treatment-specific biomarker that is associated with superior survival with taxane therapy over ARS-directed therapy in the clinical setting. As for 193 mCRPC patients whose CTCs were determined to be AR-V7-negative, Graf et al. observed that AR-V7-negative patients had superior survival on ARS inhibitors over taxanes [106]. In addition, Gupta et al. suggested that *PTEN* loss and *BRCA2* gain were associated with significantly worse outcomes in 40 AR-V7 negative mCRPC patients treated with abiraterone/enzalutamide [124]. Thus, AR-V7-negative patients are more likely to have better survival time among metastatic PCa patients treated with enzalutamide or abiraterone. Intriguingly, Hench et al. observed that the expression of AR-V7 mRNA underwent a dynamic change in the progression of mCRPC, including recovery of AR-V7 expression in six patients’ CTCs and the reversion from AR-V7-positive CTCs to either AR-V7-negtiveCTCs or “no CTCs” [125]. This conversion may be due to the inactivation of the AR signaling axis, resulting in reduced selective pressure for AR-V7 expression. Nevertheless, Belderbos et al. suggested that *AR-V7* expression in CTCs had no additional prognostic value in 127 mCRPC patients who progressed after administration of docetaxel and/or enzalutamide or abiraterone [126]. Further prospective validation is needed to explore the exact role of AR-V7 as a useful predictive biomarker in PCa patients treated with different chemotherapeutic drugs. In view of the different treatment modalities for PCa patients in the aforementioned studies, AR-V7 remains a prospective predictor of PCa. 

### 6.3. CTC Clusters

To date, only a few studies have reported the experimental acquisition and application of CTC clusters in clinical settings. These studies have shown that these clusters reflect the innate properties of primary tumors. Ortiz-Otero et al. demonstrated that CTC clusters can serve as predictive biomarkers for cancer recurrence in primary PCa. They observed that 2/15 patients experienced cancer recurrence within 2 months after primary tumor resection, and levels of individual CTCs and CTC clusters were increased in these patients at the time of surgery or after surgery. Moreover, the levels did not normalize after 2 weeks, suggesting levels of individual CTCs and CTC clusters vary considerably in PCa progression [22]. Additionally, combining CTC clusters with routine single CTC detection exhibited independent predictive value in improving prognostic stratification in mCRPC patients [127]. Furthermore, the detected CTC clusters, especially AR-V7-positive CTC clusters, are essential for the response to abiraterone and enzalutamide therapy and for predicting disease outcomes. Okegawa et al. found that 26/98 CTC cluster-positive/AR-V7-positive patients treated with abiraterone or enzalutamide had more severe bone metastasis at diagnosis, pain, higher alkaline phosphatase levels, and visceral metastases [27]. Consistently in breast cancer, larger CTC clusters have been indicative of a higher risk of patient death [14]. Therefore, CTC clusters display the ability to further stratify death risk for patients with a high level of single CTCs.

With the development of technologies such as single-cell sequencing, the understanding of CTC has become more comprehensive on a micro-level scale. However, these advanced technologies face mounting issues when they are utilized in clinical practice. For instance, single-cell sequencing, a promising method for studying PCa properties and molecular mechanisms, has revealed CTC heterogeneity on a single-cell scale. In PCa patients, the biomarkers and RNA profiling of single CTCs from each display considerable heterogeneity, including expressions of AR gene mutations and AR-V7. Moreover, tumor cells in CTC clusters also exhibit various expressions of biomarkers and genes. Therefore, research on more relevant CTCs is crucial to acquiring specific and sensitive outcomes for PCa patients. Using epithelial biomarkers (e.g., EpCAM) to identify CTCs, researchers established single-cell sequencing profiling of single CTCs that exhibit immense heterogeneity [51]. This indicates that more specific and effective selection criteria for CTCs are needed to obtain more accurate results. Thus, the CTC cluster, possessing greater metastatic potential and CSC-like characteristics, is a competitive candidate for analysis by single-cell sequencing. Research on PCa on a micro-level scale means any tiny mistake can contribute to drastically different conclusions. Researchers must stay alert for this.

## 7. Conclusions

Technical difficulty remains a challenge for the purification of CTC clusters. Recently, a variety of “label-free” enrichment technologies for CTC clusters have emerged with improved harvest efficacy [128]. However, they remain marginally satisfactory in view of weak specificity and inadequate harvesting. Therefore, the development of enrichment and isolation techniques with high acquisition efficiency and convenient downstream cellular analysis is in urgent demand.

The metastatic and pluripotent characteristics of CTC clusters are of great significance to cancer research. The formation of CTC clusters is pivotal to understanding how and why CTC clusters are heterogeneous. CTC clusters undergo EMT and contain a mix of cell phenotypes, especially the hybrid E/M phenotype, which plays various roles in metastasis. This might provide clues to a better understanding of tumor cell migration and colonization.

The CTC cluster is a promising biomarker of tumor management, yet research on its application has been insufficient. In the future, researchers will be able to study CTC clusters from the perspectives of enumeration, genome profiling, protein, and telomere. Future work on the origin, formation, and migration of CTC clusters will provide researchers with further insight into cancer biology and liquid biopsy.

## Figures and Tables

**Figure 1 cancers-14-03985-f001:**
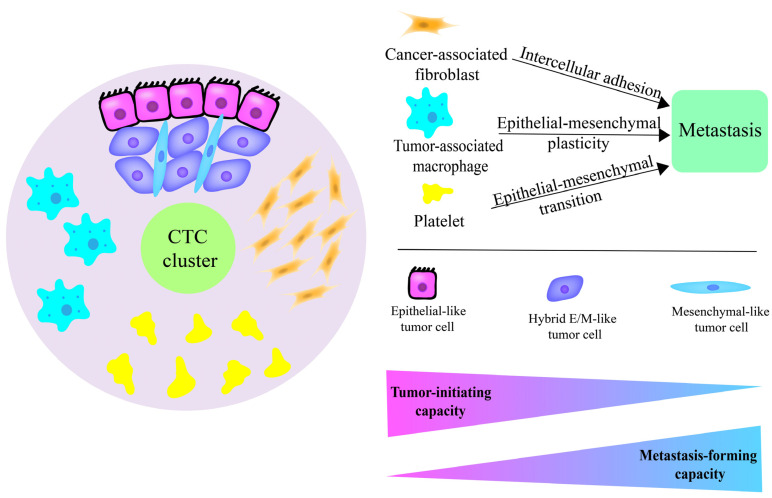
Cells that make up circulating tumor cell (CTC) clusters. CTC clusters include tumor cells and non-malignant cells that consist primarily of tumor-associated macrophages (TAMs), cancer-associated fibroblasts (CAFs), and platelets. Tumor cells undergo the epithelial–mesenchymal transition (EMT), and they exhibit three phenotypes: epithelial phenotype, mesenchymal phenotype, and hybrid E/M phenotype. The epithelial phenotype has the capacity for tumor initiation, and the mesenchymal phenotype has the capacity to form metastasis. The hybrid E/M phenotype is equipped with both capacities. TAMs, CAFs, and platelets each play roles in metastasis and prostate cancer invasion.

**Figure 2 cancers-14-03985-f002:**
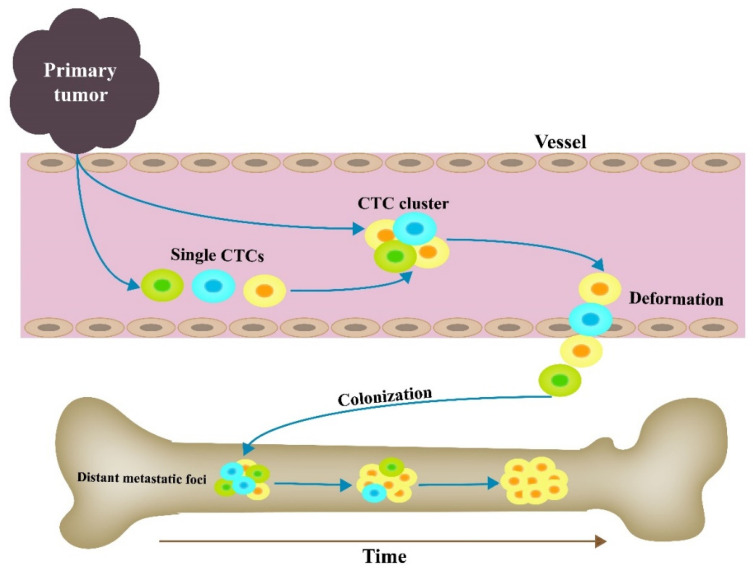
Illustrations of circulating tumor cell (CTC) cluster formation, extravasation, and colonization. CTC clusters originate from pieces of primary tumors or are assembled by single CTCs in peripheral blood. Then, CTC clusters reversibly unfold into single chains when they go through vessels, and they invade other parts of the body. Metastatic foci formed by CTC clusters are increasingly dominated by a single clonal population.

**Figure 3 cancers-14-03985-f003:**
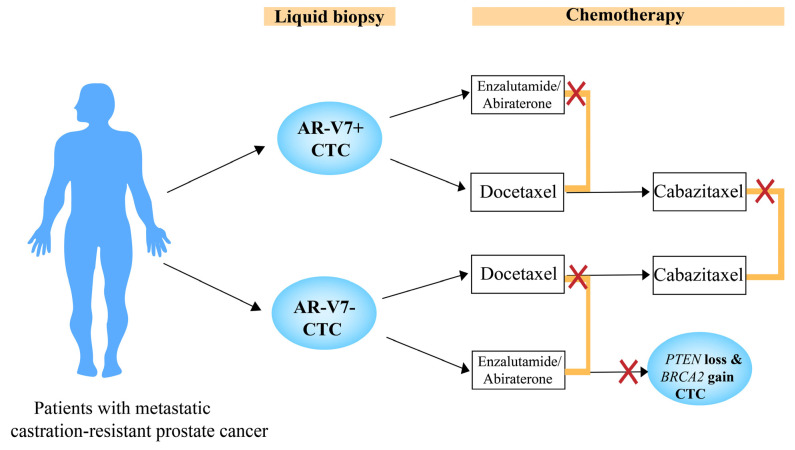
AR-V7 (androgen receptor V7) as an indicator of chemotherapy in patients with metastatic castration-resistant prostate cancer (mCRPC). mCRPC patients who have pretherapy AR-V7-positive circulating tumor cells (CTCs) are associated with poorer outcomes for enzalutamide/abiraterone over taxanes. However, mCRPC patients who have pretherapy AR-V7-negative CTCs are associated with better outcomes on enzalutamide/abiraterone; subsequently, the alterations of *PTEN* loss and *BRCA2* gain in CTCs indicate worse outcomes. In addition, cabazitaxel at a lower dose is associated with poorer outcomes after docetaxel in AR-V7-positive patients, compared to AR-V7-negative patients. Note that this figure should not be used for clinical decision-making.

**Table 1 cancers-14-03985-t001:** CTC analytes and isolation devices in prostate cancer.

Subcategory	Isolation Technology	Basis of Detection	Key Features	Ref.
Antibody	CellSearch system	EpCAM	The most widely validated CTC detection technology	[31]
Antibody	-	Cell-surface vimentin	The ability to detect CTCs undergone EMT	[32]
Gene transcripts	AdnaTest	*KLK3*, *PSMA*, and *EGFR* PCR	High sensitivity	[33]
Gene transcripts	DDPCR	*KLK2*, *KLK3*, *HOXB13*, *GRHL2*, and *FOXA1* PCR	Low blood volume, little on-site processing, and long stability for batch processing	[33]
Microfluidics	Cluster-Chip	Cell–cell adhesion	Label-free, the ability to isolate unfixed CTC clusters from unprocessed whole blood specimens	[25]

Abbreviations: CTC—circulating tumor cell; EpCAM—epithelial cellular adhesion molecule; EMT—epithelial–mesenchymal transition; PCR—polymerase chain reaction.

**Table 2 cancers-14-03985-t002:** CTC enumeration.

Cancer Type	Results (from Published Trials)
**Localized prostate cancer**	CTC in patients with localized prostate cancer had a larger number than in healthy volunteers. In patients with stage T2 tumors, the presence of Gleason pattern 5 was positively correlated with CTC positivity (rho = 0.59, *p* < 0.001) [111].
**Localized prostate cancer**	CTCs were detected in 17 patients (range: 1–clusters with >100 epithelial cells) without significant correlations to PSA levels or Gleason scores [112].
**Localized prostate cancer with ≥1 high-risk factors** (**PSA > 20 ng/mL, Gleason 8–10, stage T3–4**)**,**	CTCs were detected in 5/65 patients at diagnosis, 8/62 following neoadjuvant androgen deprivation, and 11/59 at the end of radiotherapy. Positive CTC status was not significantly associated with any clinical or pathologic factor. Detection of CTCs was not significantly associated with OS (*p* > 0.40) [113].
**Oligometastatic hormone-sensitive prostate cancer**	CTCs were detected in 51/54 patients, and M-CTCs detection rates were 67%. A positive correlation was found between the M-CTC count and number of bone metastases [118].
**Metastatic castration-resistant prostate cancer**	Higher baseline CTC count was significantly associated with worse OS, PFS and time to PSA progression [114].
**Metastatic castration-resistant prostate cancer**	In 114 metastatic castration-resistant prostate cancer patients treated with cabazitaxel, CTC counts were independently associated with PFS and OS [115].
**Metastatic castrate sensitive prostate cancer**	Patients with undetectable CTCs had nearly 9 times the odds of attaining 7-month PSA ≤ 0.2 vs. >4.0 (N = 264) and 4 times the odds of achieving > 2 years PFS (N = 336) compared to men with baseline CTCs ≥ 5 [116].

Abbreviations: CTC—circulating tumor cell; PSA—prostate-specific antigen; OS—overall survival; PFS—progression-free survival; M-CTC—mesenchymal-circulating tumor cell.

**Table 3 cancers-14-03985-t003:** AR-V7 in metastatic castration-resistant prostate cancer.

Ref.	Results (from Published Trials)
[121]	AR-V7 was detected in CTCs of 34/69 metastatic castration-resistant prostate cancer patients. AR splice variants were expressed in higher levels in CTCs than in paired extracellular vesicles.
[122]	21/37 patients CTC-positive before starting treatment with enzalutamide or abiraterone: 24% of CTC-positive patients were defined as AR-V7-positive. Positivity for each variable was significantly associated with poorer rPFS and OS.
[123]	Patients with AR-V7–positive CTCs before ARS inhibition had resistant posttherapy PSA changes, shorter rPFS, and shorter OS than those without AR-V7–positive CTCs.
[124]	Consistent *PTEN* loss and inconsistent *BRCA2* gain were associated with significantly worse outcomes in AR-V7-negative CTC patients treated with abiraterone/enzalutamide.
[125]	26/95 patients had ARFL+ARV7+, 22/95 patients had ARFL+ARV7−, 22/95 patients had ARFL−ARV7−, and 1/95 patient had ARFL−ARV7+ CTCs at baseline.
[126]	AR-V7 expression in CTCs was not associated with OS.
[102]	CTC expression of AR-V7 was significantly associated with OS. In patients treated with cabazitaxel 20 mg/sqm, median OS was shorter in AR-V7-positive than -negative patients (6.6 vs. 14 months).

Abbreviations: AR-V7—androgen-receptor splice variant 7; CTC—circulating tumor cell; AR—androgen receptor; PFS—progression-free survival; PSA—prostate-specific antigen; OS—overall survival; ARS—; rPFS—radiographical progression-free survival.

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
