# Peer review of "Insights into Circulating Tumor Cell Clusters: A Barometer for Treatment Effects and Prognosis for Prostate Cancer Patients"

_cancers, 2022, doi:10.3390/cancers14163985_

Round 1

Reviewer 1 Report

The paper by Lu et al provides a comprehensive review of the theoretical foundations of CTC clusters research and their potential clinical value and applications. This is a very interesting and promising topic in cancer. The paper is written in a very clear and sequential manner. The limitations of current studies in this research field are also properly presented and discussed.

The main disadvantage is that there are other reviews on this topic, though they do not specifically address prostate cancer. I have just a few concerns that are listed below:

1) The title is somewhat misleading because only the "Clinical applications" section focuses on Prostate Cancer. A more focused approach to prostate cancer would have made the article more interesting to read.

2) There are no references in section 2 "CTC clusters and Single CTCs". References are also missing in line 99.

Minor:

- Lines 20-21. The authors must revise/improve the first sentence of the abstract.

- Lines 93-95 should be revised because "In conclusion" appears twice.

-Line 99. An explanation of what EMT means is required for clarification.

-The quality of the figures can be improved (in particular the text embedded in the figure)

-Line 204-205. The sentence is incomplete.

-Line 281. Do you mean Figure 2?

-Line 334 revise “Due”

-Line 507 “is needed…”

Author Response

Dear reviewer:

Thanks very much for taking the time to review this manuscript. I really appreciate all your comments and suggestions! Please find my itemized responses below and my revisions/corrections in the re-submitted files.

Point 1:The title is somewhat misleading because only the "Clinical applications" section focuses on Prostate Cancer. A more focused approach to prostate cancer would have made the article more interesting to read.

Response 1: We are grateful for the suggestion. To be more in accordance with prostate cancer, we have added a more detailed interpretation regarding findings on EMT in CTCs of prostate cancer in section 3 "CTC Clusters and EMT"  in the revised manuscript. 

Point 2:There are no references in section 2 "CTC clusters and Single CTCs". References are also missing in line 99.

Response 2: We apologize for the missing references in the original manuscript. We have inserted references in section 2 "CTC clusters and Single CTCs" and the second sentence of section 3.1 "Locations of cells with different E/M states in CTC cluster and invasion"  in the revised manuscript.

Point 3: Lines 20-21. The authors must revise/improve the first sentence of the abstract.

Response 3:We agree with the comment and re-wrote the sentence in the revised manuscript as the following: Prostate cancer (PCa) exhibits high cellular heterogeneity across patients. Therefore, it is an urgent need for more real-time and accurate detection methods, in both prognosis and treatment in clinical settings. 

Point 4: Lines 93-95 should be revised because "In conclusion" appears twice.

Response 4: We apologize for the repeated sentence in the original manuscript. We have deleted the repeated sentence in the revised manuscript.

Point 5: Line 99. An explanation of what EMT means is required for clarification.

Response 5: We are grateful for the suggestion. We have added the explanation of EMT in section 3.1 "EMT and cellular locations of CTC cluster"  in the revised manuscript.

Point 6: The quality of the figures can be improved (in particular the text embedded in the figure)

Response 6: We are grateful for the suggestion. We have improved 3 figures, especially the fig.2, in the revised manuscript.

Point 7: Line 204-205. The sentence is incomplete.

Response 7: We apologize for the grammar problems in the original manuscript. We re-wrote the sentence in the revised manuscript as the following: It has been demonstrated that CTC clusters can reduce CTC apoptosis, elevate cell viability, and promote the ability to re-form clusters.

Point 8: Line 281. Do you mean Figure 2?

Response 8: We are grateful for the suggestion. Figure 1 is used to explain the role of the non-malignant cells as the metastatic supporter or promoter in CTC clusters in section "4.2. Role of CTC cluster components". We have inserted "Fig.1" in a more proper place in section 4.2 in the revised manuscript.

Point 9: Line 334 revise “Due”

Response 9: We apologize for the language problems in the original manuscript. We have revised ''Due" to due" in the revised manuscript.

Point 10: Line 507 “is needed…”

Response 10: We apologize for the grammar problems in the original manuscript. We have revised "need" to "needed" in the revised manuscript.

Reviewer 2 Report

In present review, authors have discussed the potential of prostate cancer CTCs  for better diagnostic and therapeutic use. I have several reservations, my comments are appended as below:

1. My primary concern is that authors claims the use of CTCs when metastasis is set in, in which there is already maximum damage done. Are there studies on detection of prostate cancer in primary form?

2. Devices to isolate CTCs: provide table.

3. Reference 32, 36- this seem interesting reports. Explain the cancer type.

4. Experimental systems of PC to detect CTC- provide a table.

5. Line 182-184- share the details on type of methylation.

6. Authors should provide tabular view of studies in PCa patients on CTCs, especially in settings of metastasis to distant organs including bone.

7. While referring to clinical/experimental studies, authors should always explain the model and in case of clinical studies, no of patients.

8. Line 384- explain the type of treatment.

9. CTC enumeration: provide table.

Author Response

Dear reviewer:

Thanks very much for taking the time to review this manuscript. I really appreciate all your comments and suggestions! Please find my itemized responses below and my revisions/corrections in the re-submitted files.

Point 1: authors claims the use of CTCs when metastasis is set in, in which there is already maximum damage done. Are there studies on detection of prostate cancer in primary form?

Respond 1:Thank you for your comment, and our reply is as follows:  There are several studies on CTCs in primary prostate cancer.  Some studies found an association between CTC positivity and Gleason scores in primary prostate cancer (see reference 115), but some reported that detection of CTCs was not significantly associated with clinical or pathologic factors (see reference 116, 117). Therefore, the role of CTCs in primary prostate cancer is debatable. We have discussed the CTCs of primary prostate cancer in section 6 "CTCs in Clinical Application", especially in section 6.1 "CTC enumeration".

Point 2: Devices to isolate CTCs: provide table. & Point 4: Experimental systems of PC to detect CTC- provide a table.

Respond 2:Thank you for your suggestion. On account CTC isolation and detection are closely related, we have put CTCs analytes and isolation devices added into one table in section 5 "Separation Techniques and Devices" in the revised manuscript.

Point 3: Reference 32, 36- this seem interesting reports. Explain the cancer type.

Respond 3: Thank you for your comment. Researchers used a lineage-labeled mouse model of pancreatic ductal adenocarcinoma and human breast cancer cell lines in references 32 and 36 (references 38, 44 in the revised manuscript), respectively. 

Point 5: Line 182-184- share the details on type of methylation.

Respond 5:Thank you for your comment. Researchers have reported that meta-
gene plot of CpG methylation reveals comparable methylation levels between single CTCs and CTC clusters across CpG islands, gene bodies, upstream (promoters) and downstream regions, including a drop of CpG methylation around the transcriptional start site. Then they assess overlapping regions with
a >= 70% methylation difference between single CTCs and CTC clusters, suggesting a total of 1,430 DMRs, of which 909 hypomethylated in CTC clusters and 521 hypomethylated in single CTCs.  CTC cluster hypomethylated TFBSs revealed a remarkable enrichment for stemness-related transcription factors that coordinately regulate proliferation and pluripotency, including OCT4, NANOG, SOX2, and SIN3A, paralleling embryonic stem cell (ESCs) biology. Differently, single CTCs featured hypomethylation of other TFBSs, including those that are occupied by MEF2C, JUN, MIXL1, and SHOX2, commonly enriched in various cancers. 

Point 6: Authors should provide tabular view of studies in PCa patients on CTCs, especially in settings of metastasis to distant organs including bone.

Respond 6:Thank you for your suggestion. On account the studies in prostate cancer on CTCs are mainly about the clinical application, we have added 2 tables of studies on CTC enumeration and AR-V7, respectively.

Point 7: While referring to clinical/experimental studies, authors should always explain the model and in case of clinical studies, no of patients.

Respond 7: Thanks for your suggestion a lot. We have explained the models and the number of patients in studies in the revised manuscript.

Point 8: Line 384- explain the type of treatment.

Respond 8: Thank you for your comment.  CTCs are helpful to reflect the chemotherapy of PCa patients. For example, high ARV7 in 22RV1DR and LNCaP-ARV7 cells correlated with taxane resistance, suggesting AR-V7 has a predictive role in the taxane response (see reference 112 in the revised manuscript).  What's more, a study demonstrated the property of CTCs to be a source of phenotypic information in relation to therapy response to the metastatic PCa (see reference 113 in the revised manuscript).  In addition, a  man with prostate cancer and metastasis of the pubic bone underwent neoadjuvant androgen deprivation and docetaxel therapy. Biopsy specimens are difficult to obtain and might not reflect the precise extent of the disease owing to heterogeneity in patients with CRPC. Thus, researchers performed a liquid biopsy to isolate CTCs, and overall 156 CTCs were detected per 7.5 mL. Almost all CTCs were androgen receptor-negative in the nucleus. They diagnosed the five nodules as lung metastases from docetaxel-resistant CRPC with few AR-signaling-dependent cancer cells. The patient was initiated on CBZ chemotherapy instead of using a second-generation AR-targeting agent. After 2 cycles of CBZ chemotherapy, PSA level decreased to < 0.01 ng/mL and the lung metastases completely disappeared, with a reduced CTC count of < 5.

Point 9: CTC enumeration: provide table.

Respond 9: Thank you for your suggestion. We have added a table of CTC enumeration in section 6.1 " CTC enumeration" in the revised manuscript.

Reviewer 3 Report

In this review, the authors summarized the role of CTC clusters in prostate cancer metastasis and clinical applications. The topic is novel, and the text is nicely written. It is suitable for publication in the current format. There is a minor suggestion. It will make the conclusion clearer if authors consider dividing the part “3 CTC Clusters and EMT” as a subsection of part “4 CTC Clusters and Metastasis”. 

Author Response

Dear reviewer:

Thanks very much for taking the time to review this manuscript. I really appreciate all your comments and suggestions! Please find my itemized responses below and my revisions/corrections in the re-submitted files.

Point 1: It will make the conclusion clearer if authors consider dividing the part “3 CTC Clusters and EMT” as a subsection of part “4 CTC Clusters and Metastasis”. 

Respond 1: Thank you for your suggestion. Considering EMT is important to CTCs for not only metastasis but also other functions, we have re-written the subtitles in section 3 “ CTC Clusters and EMT” to make the tighter association between EMT in CTCs and metastasis for a clearer conclusion for keep section 3 as an important part.

Round 2

Reviewer 2 Report

All my concerns are addressed.